biomechanics

elbow, knee, quadruped, gait, posture, work

**Author for correspondence:**
James R. Usherwood
e-mail: jusherwood@rvc.ac.uk

# Limb work and joint work minimization reveal an energetic benefit to the elbows-back, knees-forward limb design in parasagittal quadrupeds

James R. Usherwood[1] and Michael C. Granatosky[2]

[1]Structure and Motion Lab., The Royal Veterinary College, North Mymms, Hatfield, Herts AL9 7TA, UK
[2]Department of Anatomy, New York Institute of Technology, Old Westbury, New York, NY 11568, USA

JRU, 0000-0001-8794-4677; MCG, 0000-0002-6465-5386

Quadrupedal animal locomotion is energetically costly. We explore two forms of mechanical work that may be relevant in imposing these physiological demands. Limb work, due to the forces and velocities between the stance foot and the centre of mass, could theoretically be zero given vertical limb forces and horizontal centre of mass path. To prevent pitching, skewed vertical force profiles would then be required, with forelimb forces high in late stance and hindlimb forces high in early stance. By contrast, joint work—the positive mechanical work performed by the limb joints—would be reduced with forces directed through the hip or shoulder joints. Measured quadruped kinetics show features consistent with compromised reduction of both forms of work, suggesting some degree of, but not perfect, inter-joint energy transfer. The elbows-back, knees-forward design reduces the joint work demand of a low limb-work, skewed, vertical force profile. This geometry allows periods of high force to be supported when the distal segment is near vertical, imposing low moments about the elbow or knee, while the shoulder or hip avoids high joint power despite high moments because the proximal segment barely rotates—translation over this period is due to rotation of the distal segment.

## 1. Introduction

Theoretical consideration [1] and computer optimization [2] of bipedal gaits indicate that there is an inevitable cost to locomoting with finite step lengths in terms of limb mechanical work, and that this cost is minimized with some form of vaulting 'inverted pendulum' walking gait at low speeds, and a 'bouncing', running gait at higher speeds. However, these conclusions are not directly applicable to quadrupedal gaits.

In bipeds (humans, birds, rodent and marsupial hoppers) with zero, or at least finite, pitch moment of inertia, limb forces that do not act in line from ground contact to centre of mass impose pitch accelerations with detrimental energetic consequences. Axially loaded limbs—legs with force vectors passing along the line of the leg, close to the joint centres and broadly towards the hip—may allow passive vaulting during the stance phase of walking, but step–step transitions require work [3,4]. In running, axially loaded limbs slow the body both horizontally and vertically until the centre of mass is relatively low and the supporting leg flexed, requiring work to re-accelerate and lift the body during the second half of stance.

By contrast, the requirement of broadly axial limb loading due to pitch avoidance does not apply to quadrupeds: their shoulders and hips are located fore and aft of the centre of mass, so pitching moments applied from one end can be countered by the other without resulting in pitch accelerations.

Theoretical force profiles and foot contact timings that result in zero vertical displacements of the centre of mass, zero horizontal forces and zero roll or pitch accelerations have been described [5], and are broadly consistent with observed timings and forces of tortoises [5,6]. While the original intent of such modelling was in exploring strategies that prevent excessive roll and pitch despite strides of relatively very long duration, it also demonstrates a strategy that allows weight support and progression while approaching absolutely zero limb work [6]. The term 'limb work' here is the mechanical work associated with forces and deflections from ground contact (the foot), all the way to the centre of mass, for each leg—it does not include work associated with accelerating the limb mass.

Body weight *could* be continuously and exactly supported with purely vertical forces from the limbs. Hips and shoulders *could*, therefore, translate the centre of mass perfectly horizontally and, if horizontal forces are avoided, no limb power is ever required. Force profiles of walking alligators and tortoises appear highly consistent with this wheel-like or 'sliding' strategy [6] (this is not typical for mammals, particularly the familiar, larger species—see below). Note that zero limb work as defined here, while providing one option for demanding zero work from the muscle, is not the only one. For instance, with hypothetical perfect elastic mechanisms, or suitable transfer between centre of mass and rotational energies (as in a double pendulum that may be a helpful analogy for a swinging gibbon), limb force vectors and centre of mass velocities are not continuously perpendicular, limb work is performed, but no energy need necessarily be dissipated or generated from the muscle. More importantly, finding a limb performing low limb work would not be sufficient to suggest low work from the muscle: if low limb power is achieved using simultaneous positive and negative muscle powers, there may still be a high, costly, muscle power demand.

Principles that might allow the zero limb-work strategy to be achieved with low muscular work have recently been described with analogy to linkages of the industrial revolution [6]. Sprawled limbs with predominantly vertical axis joints allow translation without joint work; by contrast, parasagittal limbs with predominantly transverse-axis joints must experience simultaneous and cancelling positive and negative joint powers, but this could be achieved with appropriate passive linkages [6]. Joint power and joint work here are those associated with moments and angular deflections or velocities about each joint. Inter-joint energy transfer, often due to muscle–tendon units crossing at least two joints, has long been recognized [7], inhibiting simple calculation of functionally relevant work from merely summing joint works. The mechanical work required from limb muscles in order to support the body mass during translation may, therefore, relate to some intermediate between limb and joint works. It is important to note that these demands on muscle are not the only sources of physiological cost: work and power associated with accelerating limb masses for protraction and retraction may be significant; as may be the costs associated with activating muscle isometrically. However, weight support during travel with low mechanical work demand on muscle—whatever the other sources of physiological cost—is presumably an important feature of economical locomotion, and traits that facilitate this (in addition to tendon elastic energy storage and recoil [8]) may provide insight into animal form and function.

## 2. Paper scope

This paper aims to (i) demonstrate the contrasting demands for limb work and joint work minimization; (ii) provide evidence that quadrupeds generally show features consistent with a compromise between the two forms of work minimization; and (iii) show how the elbows-back, knees-forward limb geometry can act to reduce the joint work demands of a low limb-work stance.

### (a) Axial forces reduce joint work not limb work

The existence of decelerating and accelerating impulses in running or trotting—with their associated energetic costs—does not appear consistent with simple limb-work minimization. While a point mass model of bipedal running demands that limb forces must be maintained axially (along the line of the leg from foot through centre of mass), the same is not true for trotting. Here (figure 1), we model (details in the electronic supplementary material) above-walking-speed gaits with a half-sine vertical force profile sufficient to oppose body weight (a reasonable approximation for running and trotting [9,10]) and include horizontal forces such that ground reaction force (GRF) vectors are continuously orientated a proportion $p$ between vertical and upper (hip or shoulder) joint. Limb powers are calculated as the dot product of GRF and centre of mass velocity. Joint powers are calculated using the force vectors and the geometry of (i) an isolated two-joint forelimb with backward-angled 'elbow' or (ii) an isolated hindlimb with forward-angled 'knee' joint (figure 1). The instantaneous power at a joint is the product of the moment created by the GRF vector and the angular velocity of the joint at that instant. Limb and joint works are the integrated respective positive powers over the stance. Leg (assumed to be massless) geometries are determined here with equal length upper (thigh/upper arm) and lower (shank/forearm) elements, though results are not sensitive to this. Initial and final height and horizontal displacement from midstance are assumed to be equal, resulting in a symmetrical path of the proximal joint about midstance. The model is intended to be generic, so absolute lengths, speeds and timings are unimportant (powers are represented in arbitrary units); however, the geometry is broadly appropriate for a human (1 m leg length) sprinting at $8\,\mathrm{m\,s^{-1}}$ with both stance and aerial duration of 0.125 s, landing with a slightly flexed leg (initial start height of 0.85 m). Leg flexion/extension is prescribed by the centre of mass trajectory and the feasible geometry of the two-segment leg.

The simulation confirms that, for a running (trotting or hopping)-style gait, the vertical force strategy ($p = 0$) demands less limb work than the axially loaded limb [11]. This can be explained as, with purely vertical forces, only the work required to accelerate the hip (/shoulder) up from its lowest point is required; any deviation from vertical forces demands an additional limb work to accelerate the hip (/shoulder) horizontally.

By contrast, *joint* work is low if the forces are orientated directly towards the proximal hip or shoulder joint ($p = 1$); purely vertical forces result in very high joint powers (figure 1). The general principle can be explained if (i) all force orientation strategies use approximately consistent kinematics, such that each joint has broadly consistent angular profiles across options, but (ii) GRF vectors passing close to joint

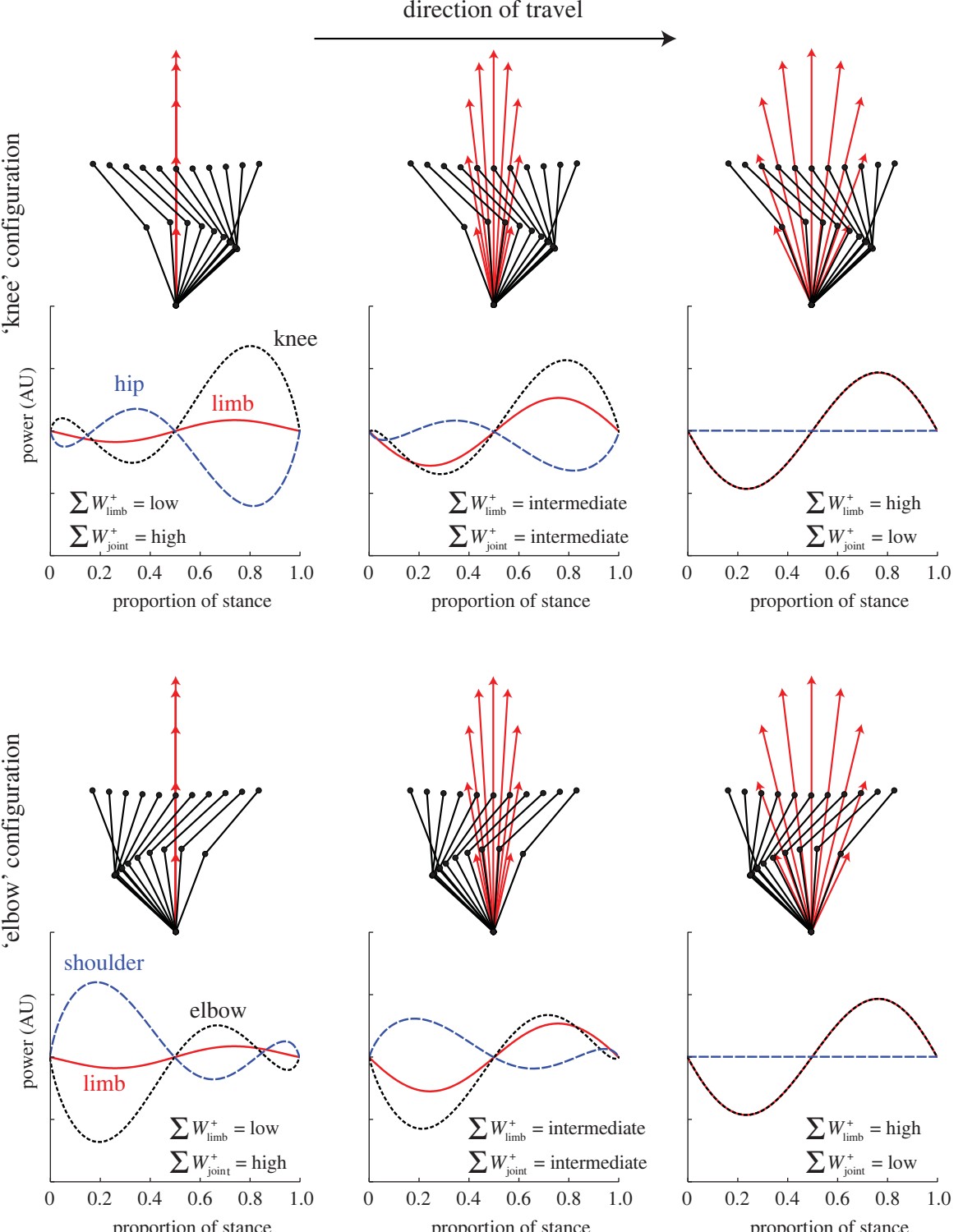

**Figure 1.** The consequences of orienting force vectors vertically ($p = 0$, first column), half-way between proximal joint and vertically ($p = 0.5$, second column) and directly through the proximal joint ($p = 1$, third column). Stick cartoons with red arrows show proximal (hip or shoulder) and distal (knee or elbow) joint positions and force vectors at evenly spaced instances through stance. In this case, summed positive joint and limb work is not dependent on knee/elbow orientation (forward- or rearward-facing), though joint power profiles do differ. Directing force vectors continuously vertically minimizes limb work but results in high joint work. Force vectors maintained axially, through the proximal joint, minimize joint work but result in high limb work. (Online version in colour.)

centres result in low aggregate joint moments and therefore joint powers. Purely vertical GRFs result in large moment arms about the joints and high negative and positive joint powers. Summing the positive powers and neglecting energy transfer between joints results in high joint work costs. With symmetrical, unskewed force profiles, there is no difference between elbow- or knee-orientation summed joint works, though they do exhibit contrasting joint power profiles which will become relevant later in this paper.

Forces directed more vertically than axially indicate a pressure towards limb work rather than joint work minimization. Were this to be observed, it would suggest—but not explicitly demonstrate—at least some degree of inter-joint power transfer. Measured vertical force profiles provide evidence of more-vertical-than-axial limb loading. It has long been recognized [11] that vertical forces would demand lower work requirements than axial, but also identified that this is not consistent with observed force orientation in

Proc. R. Soc. B 287: 20201517

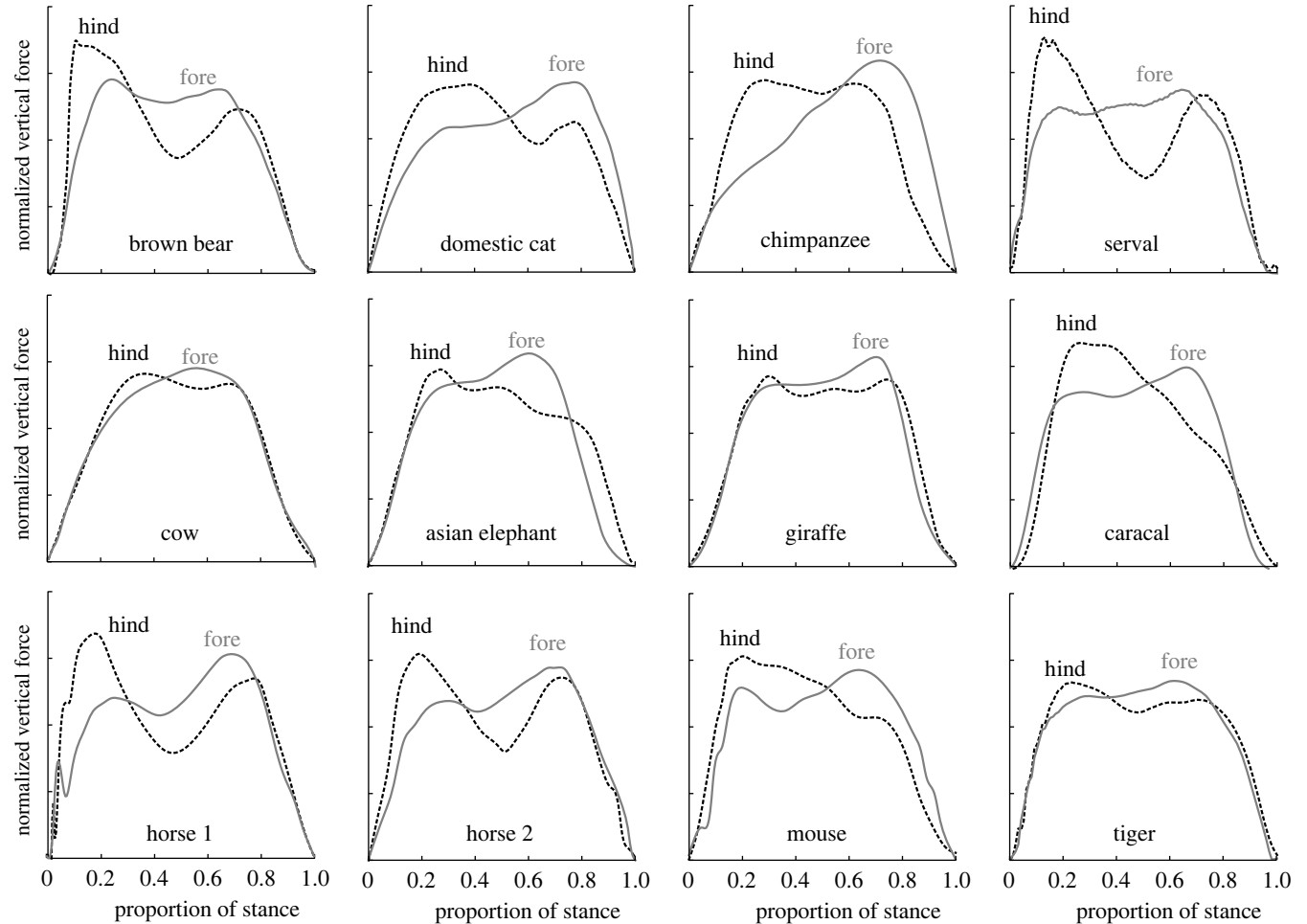

**Figure 2.** Demonstration of skews and 'M's in vertical GRF for a range of walking mammals from a range of studies and labs. Forces are normalized by mean force over stance (the area under each curve is made constant) to highlight contrasting skews. Note that this means that magnitudes of hind and fore forces are not directly comparable. Hindlimb forces tend to be skewed high in early stance; forelimb forces skewed high in late stance. Bear: [28]; cat: [29]; chimpanzee: [30]; cow: average of 143 measurements of 43 individuals (RVC); elephant: [31]; giraffe: [32]; horse 1: [33]; horse 2: [34]; mouse: [35]; serval, caracal, tiger: [36]. (Online version in colour.)

walking and running animals. This was attributed to an absence—or at least insufficiency—of multi-joint linkages to provide inter-joint power transfer [11], meaning that minimization of joint work was taken as a suitable initial cost function to consider when exploring animal limb design (though see also [12]). Minimization of joint work does appear effective in accounting for some features of bipedal [13] and quadrupedal [14,15] gait kinetics [1]. But force vectors in bipeds (humans: [16]; birds: [17]; wallabies: [18]) and quadrupeds [19] are consistently observed to be orientated between axial and vertical. If joint work was indeed the suitable energetic cost, some non-energetic explanation is then required. Stability is one such consideration [16] and has driven considerable interest, not least due to the very demanding challenges of achieving stability in bipedal robots. However, the stability issue does not appear to translate to quadrupeds: with a suitable pitch moment of inertia, stability in pitch is easily achieved [20]. In quadrupeds at least, then, we interpret orientation of force vectors between axial and vertical as indicating a compromise between joint work (axial) and limb work (vertical) minimization.

Vertical forces that are higher in early stance than late may be termed early-skewed. These are widely observed in bipeds, including young children, human sprinters and birds. Potential explanations for this range from energetic (children: [21]; birds: [22]) to stability [23] to anatomical

(sprinters: [24]; birds: [17,25–27]). Whatever account for early skew is favoured for bipeds, late skew—a common feature of quadruped forelimb vertical force profiles (figure 2)—is not consistent with any of the accounts for skew in bipeds. Instead, it points to pitch avoidance with a limb loading that is more vertical than purely axial [5], a feature of low limb-work gaits [6]. Vertical forces from the forelimb in early stance act on a large moment arm about the centre of mass, which reduces over the duration of stance; in order to impose both a net-zero pitching moment and constant weight support, this requires a vertical force profile that increases through stance, accounting for the late skew.

Such skews (not only hind-early but also and especially fore–late) are widespread across walking quadrupedal mammals (figure 2). Further, timing of peak vertical force as a proportion of stance acts as a convenient proxy to skew; this is relatively early for hindlimbs and late for forelimbs across quadrupedal tetrapods (figure 3), suggesting pressure towards low limb-work kinetics quite generally.

## (b) Vaulting reduces joint work not limb work

Limb work during weight support can theoretically be zero if purely vertical forces constantly summing to oppose body weight can be realized. While some form of vaulting may be limb-work minimizing for bipeds due to costs associated

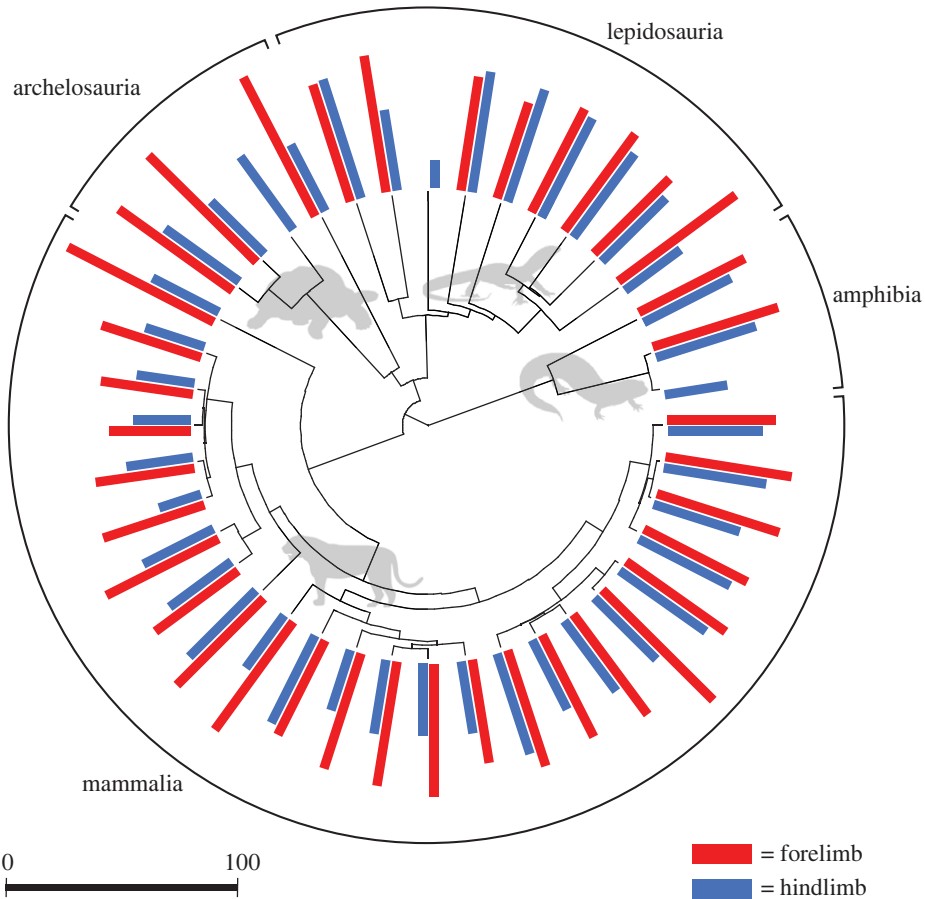

**Figure 3.** Phylogeny of species and bar graphs showing the timing of peak vertical force as a percentage of stance phase for the forelimb (red) and hindlimb (blue). Use electronic supplementary material, figure S1 as a reference for all scientific names. Across almost all quadrupedal tetrapods sampled, peak vertical force occurs later in stance phase in the forelimb compared to the hindlimb (phylogenetic paired *t*-test: *t* = 2.64, *p*-value = 0.01; [37]). It should be noted that some more basal lepidosaurs do not follow the general quadrupedal pattern, but further taxonomic sampling is required to assess whether there is any biological consistency to this finding. Data for the timing of peak vertical force in the hindlimb was reported by Granatosky *et al.* [38]. Data for the timing of peak vertical force in the forelimb previously unpublished (see electronic supplementary material, table S1). For certain species, no limb loading data were available for the forelimb. Tree topology and the depth of each node are based on the recent supertree reported at www.timetree.org [39]. (Online version in colour.)

with pitching (demanding broadly axial limb forces), the same is not the case for quadrupeds as pitch moments can be resolved through skewed force profiles. The parsimonious account for limb vectors not being constantly vertical therefore presumably falls elsewhere. Let us take the extreme case, that between-joint energy transfer is either impossible or extremely costly. Might this account for vaulting 'inverted pendulum' walking gaits?

The geometry of a symmetrical arcing-vault stance phase of a single limb operating at duty factor (proportion of stride each foot is in contact with the ground) DF = 0.5 allows the mechanical works associated with joints to be approximated. In the case of a quadruped walking with relatively short steps, with hips and shoulders relatively far from the centre of mass, issues surrounding pitch can be neglected. In this case, the strategy for limb work minimization is to maintain a constant, horizontal hip (/shoulder) joint trajectory with constant vertical force and zero fore–aft forces. But what would this demand in terms of joint power for a parasagittal leg? This strategy would require flexion of one or more joints (predominantly the knee or elbow) while under body weight load; also, as horizontal forces are to be avoided, moments about the upper joint (hip or shoulder) are required. In the case of exactly constant zero *limb* power, the positive and negative *joint* powers coming from upper and lower joints continuously cancel: the

knee flexes, absorbing energy during the first half of stance, while the hip contributes positive work; the knee extends providing positive joint power during the second half of stance, while the hip dissipates. For a symmetrical stance, the minimum joint work $W_{joint}^{+}$ to maintain the zero limb work strategy is simply double the work dissipated by the knee $W_{knee}^{-}$ (or elbow):

$$\sum W_{joint}^{+} = 2W_{knee}^{-}. \tag{2.1}$$

The negative work performed by the knee as it compresses to midstance depends on the vertical force (constantly opposing weight) and the limb compression as it deviates from an arc over a foot contact about an angle $\phi$. Up to midstance,

$$W_{knee}^{-} = mg(L_0 - L_0 \cos(\sin^{-1}(\phi/2))), \tag{2.2}$$

where *mg* here is the weight being supported by the (hind or fore) pair of legs and $L_0$ the initial leg length. The summed joint work for the horizontal, zero limb work strategy, over the entire stance, is therefore (using small angle assumptions) approximately:

$$\sum W_{joint}^{+} = 2W_{knee}^{-} = 2mgL_0\left(1 - \left(1 - \frac{\phi^2}{4}\right)\right) = \frac{mgL_0\phi^2}{2}. \tag{2.3}$$

The alternative extreme strategy is to vault with a curvature resulting in an arc of radius $L_0$. This would require zero knee flexion, purely axial limb forces, and be achievable passively over the majority of stance, but would result in work during the step transition. Adopting the collisional analysis approach [3,4], the work demanded from such a transition can be approximated for small angles by

$$W_{transition} = \frac{JmV^2\phi^2}{2},\qquad(2.4)$$

where the term $J$ is a 'collision reduction factor' [4] and is dependent on the details of step transition geometry. A low value ($J = 0.25$) would be achieved with a very brief, high-force impulse perpendicular to the path of the supported mass at the end of the vault, timed immediately before transition. Something approximating this strategy—effectively the push-off at the end of stance familiar to walking adult humans—accounts for the second hump in M-shaped vertical ground reaction forces; the first hump is associated with the rapid deceleration following heelstrike. With the idealized 'inverted pendulum' strategy of walking (whatever the value of $J$), no work is performed during the vault, during which axial forces are continuously perpendicular to the hip joint velocity. The summed positive joint work is thus the same as the limb work—that associated with the step transition.

It is convenient to express velocity in a non-dimensional form $\hat{V} = V/\sqrt{gL_0}$ to allow comparison across sizes (see [40], though note that this form is the square root of their version of Froude number). With this normalization, the human walk-run transition typical occurs at close to $\hat{V} = 0.7$. Substituting non-dimensional velocity for velocity gives the condition for which the extreme vaulting strategy results in lower joint work than the horizontal-path, zero limb-work strategy:

$$\frac{Jm\hat{V}^2 gL_0\phi^2}{2} < \frac{mgL_0\phi^2}{2}\qquad(2.5)$$

and

$$\hat{V} < \sqrt{\frac{1}{2J}}.\qquad(2.6)$$

With no collision reduction, the vaulting strategy requires lower joint work than the constant horizontal strategy up to $\hat{V} \approx 0.7$; the transition occurs at higher (relative) velocities with more effective (lower $J$) collision reduction mechanisms. This comparison demonstrates that vaulting reduces joint work—at the cost of above-zero limb work—at low speeds and/or with effective collision reduction. (Note that only vaulting and zero work extremes are considered; an intermediate strategy presumably minimizes joint work at close to the $\hat{V}$ limit.) Vaulting with some degree of collision reduction can be inferred from 'M-shaped' vertical ground reaction forces, though we should note that models treating the leg as an obligate spring can also achieve 'M' profiles [23]. Within the vaulting paradigm, forces close to midstance become relatively low due to the centripetal acceleration of the arcing mass [41], while the 'shove' at the end of stance and 'crash' at the beginning is consistent with the collision minimizing strategy described for walking bipeds [3,4].

Among energetic (as opposed to spring-dominated) accounts, minimization of joint work rather than limb work is therefore required in order to account for the M-shape of vertical force profiles for quadrupeds, a feature observed

**Table 1.** Contrasting force requirements for limb-work and joint-work minimization while avoiding high pitching moments in quadrupeds. Vertical force profiles of walking mammals are generally consistent with a compromise between the two, with some degree of 'M' indicating vaulting and collision-reduction consistent with joint-work minimization, and some degree of skew (early hind, late fore) consistent with pitch-avoidance despite more-vertical-than-axial limb forces, a feature of limb work minimization.

|  | limb-work minimizing | joint-work minimizing |
| --- | --- | --- |
| orientation of limb forces | vertical; zero horizontal | axial; horizontal forces vary with limb angle |
| vertical force profile | sawtooth/modified sawtooth. Early skew (hind); late skew (fore) | M-shaped (vaulting) symmetrical (trotting) |

quite widely across walking mammals (figure 2), to a greater or lesser extent depending on speed. Table 1 summarizes the contrasting features of limb force profiles relating to limb work versus joint work minimization.

## (c) Low limb work combined with low joint work is facilitated by elbows-back, knees-forward posture

If we accept that parasagittal quadrupeds (predominantly mammals, but consider also chameleons and some dinosaurs) experience costs associated with both limb and joint works, how might these costs be minimized? And what principles might be discerned concerning the limb structure? Given both the vast potential parameter space of joint kinematics and limb kinetics, the limited understanding of the relative weighting of limb- and joint-associated costs, and the diversity of animal form to be considered, we limit ourselves to a highly reductionist simplification of animal legs. Here, we consider the forelimb and hindlimb, each consisting of two segments connected by an 'elbow' or 'knee', respectively. In reality, depending on level of biological realism demanded, there may be one [12,42] or multiple additional segments, but here we take advantage of the geometric constraints of only two segments per limb in order to consider the implications of forward- or backward-pointing elbows and knees: any position of proximal joint (hip/shoulder) with respect to the foot and located closer than one leg length from the foot can be achieved with only two feasible positions of the elbow/knee—'forward' or 'backward' (figure 4).

In order to determine the consequences of elbow/knee orientation in terms of joint power for limb kinetics achieving low limb power, and also simplifying the analysis, we calculate the joint powers required to support force profiles that result in zero limb power (simulation details in the electronic supplementary material). Conveniently, this means that only the moment (from the vertical force and horizontal distance of the foot from the proximal joint) and the angular velocity of the proximal segment need to be calculated: the product of these gives the hip or shoulder joint power, and the knee or

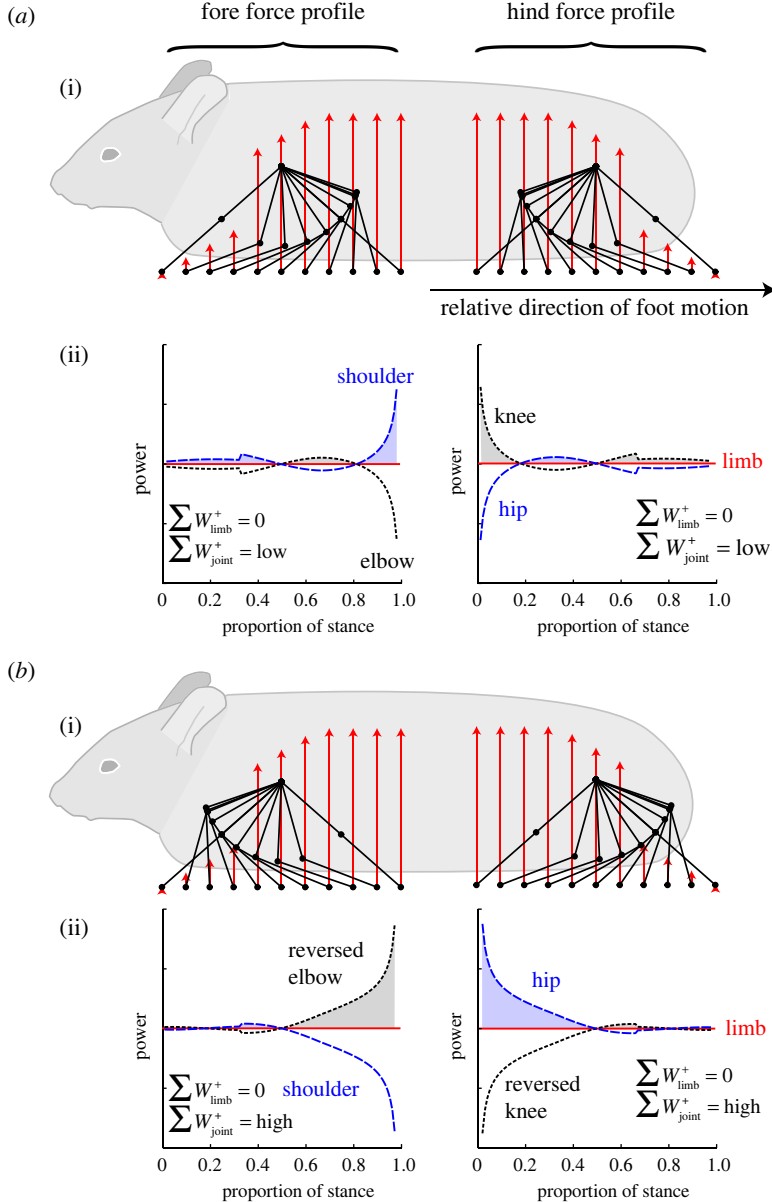

**Figure 4.** The consequences of elbow and knee orientation on joint powers and joint work while supporting weight with a force profile resulting in zero limb work. The skew in vertical force profile (red arrows show force vectors), with the forelimb supporting a greater proportion of body weight in late stance, favours the rear-pointing elbow configuration because periods of high force are supported with either low moment arm (about the elbow, as the distal segment is approximately vertical) or low angular velocity (about the shoulder, as horizontal motion is predominantly due to rotation of the vertical distal limb element). The forward-orientated knee is similarly favoured for the hindlimb, as the skew in forces is reversed, with high forces in early stance. (Online version in colour.)

elbow power is simply the negative of this value as the limb power is the sum of proximal and mid-leg joint powers and continuously adds to zero. This allows the joint power profiles to be determined (figure 4) given modelled two-segment limb kinematics and theoretical force profiles preventing limb power (the special case of DF = 0.75, phase = 25% profile makes this continuously determinate [5]). The full extent of the potential stance, from straight-leg to straight-leg is modelled, though we acknowledge that this is biologically unrealistic.

## 3. Results and explanation

Unlike the case for symmetrical force profiles (figure 1), skewed force profiles result not only in different power profiles but also different joint work contributions (shaded areas, figure 4). With skewed forces, elbow or knee

orientation *does* have a bearing on the cumulative joint work. Further, given the low-to-high skew of forelimb and high-to-low skew of hindlimb vertical force profiles attributable to limb work minimization and pitch avoidance [6] and generally observed among walking quadrupedal tetrapods (figures 2 and 3), the orientation minimizing joint work for parasagittal legs is as usually observed in mammals in nature: elbows-pointing backwards and knees-pointing forwards. The physics underlying this phenomenon can be explained with simple geometrical principles, indicating this to be a general mechanism relevant to support with skewed vertical forces, and not limited solely to the special case of perfect zero-limb-work skew. For the normally orientated backwards-pointing elbow or forward-pointed knee, the higher forces are supported when the distal segment is close to vertical, resulting in small moment arms and so low elbow or knee power. Over this period, the large moment arms about the shoulder or hip coincide with high

vertical forces, but fail to result in high shoulder or hip joint powers because the proximal segment rotates at low angular velocity; translation is achieved predominantly due to rotations of the near-vertical distal segment. By contrast, counter-factual elbow and knee orientations (figure 4*b*) impose the greatest loads at instances of high angular velocity *and* high moment arm of shoulder and hip, resulting in higher summed positive joint work demand.

## 4. General conclusion

Evidence that gait kinetics of many parasagittal quadrupeds appears to be influenced by both limb and joint power reduction, and the geometric development considering strategies to reduce both, point to a novel, energetic account for the elbows-back, knees-forward geometry of parasagittal quadrupeds. A reverse accounting—that the elbow and knee orientations drive the contrasting skews in vertical force—may be plausible, but would not easily account for the distribution of skews among non-parasagittal, sprawled tetrapods (figure 3).

The caveats concerning deviation from reality (especially neglecting the action of the shoulder in the forelimb or ankle–toe segment of the hindlimb) should be remembered. The modelling and interpretation presented here relies on a two-segment reduction of quadruped legs; it remains to be determined whether the same principles apply to multi-segment limbs (bear in mind that horse legs might be considered as consisting of at least seven segments, with seven or more joints!). Also, the current model is parasagittal and planar; no real animal limb operates in precisely such a manner. Further, alternative hypotheses relating to the implications of elbow/knee orientation concerning directional stability [43] or passive release of elastic energy facilitating protraction [44] should not be dismissed, especially for highly cursorial quadrupeds. And multi-segment limbs may well offer multiple advantages, from facilitating elastic storage and recoil (see [13]) with implications in terms of passive joint stabilization (Seyfarth *et al.*, 2001 [45]). However, the benefit proposed here of reduction in joint work given a generally low limb-work force profile appears potentially very general, and applicable to parasagittal quadrupeds of diverse scales, evolutionary backgrounds (consider chameleon), and even orientations (remember sloth). Whether the phenomenon should be considered causal—'why' mammals and similar quadrupeds have backward elbows—may have to remain the realm of conjecture. As ever, a single trait may be associated with several potential advantages, these may often not be mutually exclusive, and the fossil record does not always provide evidence concerning the selective pressures being mostly keenly applied at transitional stages. But low limb and joint work costs for supporting body weight during locomotion would appear a likely selection pressure, and this is improved with the elbows-back, knees-forward limb geometry.

Ethics. See source publications where appropriate for ethics statements concerning animal studies.

Data accessibility. Data sourced from published works, provided in electronic supplementary material. Data and code available from the Dryad Digital Repository: https://doi.org/10.5061/dryad.76hdr7sv4 [46].

Authors' contributions. Conception: J.R.U.; model development: J.R.U.; acquisition and processing of data: M.C.G. and J.R.U.; initial drafting: J.R.U.; critical revision and redrafting: M.C.G. and J.R.U.

Competing interests. We declare no competing interests.

Funding. This work was supported by the Wellcome Trust (J.R.U.: [202854/Z/16/Z]).

Acknowledgements. We are grateful for the interest and support from the RVC Structure and Motion Lab., and for the various published and previously unpublished force data used in figure 2. This collaboration was initiated at a meeting of the Society for Integrative and Comparative Biology.

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
