## [Reviewer comments · Proceedings of the Royal Society B: Biological Sciences]

Review History

RSPB-2020-1517.R0 (Original submission)

Review form: Reviewer 1

Recommendation

Major revision is needed (please make suggestions in comments)

Scientific importance: Is the manuscript an original and important contribution to its field?

Acceptable

General interest: Is the paper of sufficient general interest?

Good

Quality of the paper: Is the overall quality of the paper suitable?

Good

Is the length of the paper justified?

Yes

Should the paper be seen by a specialist statistical reviewer?

No

Do you have any concerns about statistical analyses in this paper? If so, please specify them explicitly in your report.

No

It is a condition of publication that authors make their supporting data, code and materials available - either as supplementary material or hosted in an external repository. Please rate, if applicable, the supporting data on the following criteria.

Is it accessible?

Yes

Is it clear?

No

Is it adequate?

No

Do you have any ethical concerns with this paper?

No

Comments to the Author

The authors build a nice history around a very simplified model and blend out important literature that already discussed about more vertical forces from trunk balance point of view e.g. (Aminiaghdam et al., 2017; Andrada et al., 2014; Blickhan et al., 2015; Maus et al., 2010). There exist also more literature about leg segmentation (Gunther et al., 2004; Seyfarth et al., 2001; Witte et al., 2001). Nyakatura and colleagues also used joint power to reconstruct walking gaits of salamanders, and of the extinct Orabates (Nyakatura et al., 2019). Knowing that literature, the results are so novel as the authors claim. Still the authors have a point, and if they go out of their very enclosed point of view, the manuscript can be an interesting lecture for biologist that start studying leg design.

Line 59; The authors start the sentence with „in bipeds“, however, they talk only about humans. The explanation that follows about pitch moment of inertia is not valid for birds. The authors know for sure that most bipeds are birds. The authors avoid here a lot of literature that already noted the balance of trunk necessitates more vertical ground reaction forces.

Line 84 This is just valid if you have a wheel or the motion is quasi-static. I think the authors should be fair here and state this. No mammal uses the strategy proposed by the authors, as all of them display fore-aft forces and contact time are not infinitesimal.

Section: Axial forces reduce joint work not limb work

I think the model is not clear enough described by the authors. Is the model dimensionless? If not what are the contact times?

I guess the motion of the two segments is constrained by the contraction/extension of the effective leg? If so, explain where the motion of the effective leg came from? From a rat?

Why the authors use a two-segment model? Of course, I understand that the kinematic of the segments is given (but not for every reader), but to me every simplification of reality must be explained and contextualized. For me is not enough to say we know there are others that used 3 segments, but we use 2. In addition

If you are going to build a robot, I would see your point, but here you are trying to explain form and function. Well, if you have two segments, the kinematic of the segments is different to the real segment kinematic. Thus, I need that you show that the power deviations relative to a 3/4 segment leg do not alter your point.

Lines 174/180: This explanation is completely sound for the simplified model proposed by the authors. However, it is not for a leg with more than 2 segments. If you have three segments you can reduce joint excursions and therefore reduce joint work. This is the way that most animals do. For example, dogs joint elbow/knee angles do not or vary little during stance. If you have two segments, joint angles will always display important excursions and therefore higher joint works. Line 183 Well I do not agree, there are important literature showing that more vertical GRF are needed for trunk balance e.g. (Andrada et al., 2014; Blickhan et al., 2015; Maus et al., 2010). And that the skewed GRF to is related to the shift of the COM with respect to the pelvis (Aminiaghdam et al., 2017; Andrada et al., 2014).

Line 194 this is because the most proximal joint is located cranially to the COM. Following Aminiaghdam et al. 2017 a person walking prone backwards should display the late-skew. About bipedy and skewed GRFs. The bipedal SLIP model produces both asymmetries. Left-skewed is self-stable right-skewed is not.

Line 318 There is another explanation for the M-shape of the GRF. While the inverted pendulum and collision models cannot reproduce the shape of the GRF, the SLIP model does (Geyer et al., 2006). Again, the authors seem to blend out literature that do not agree with their claims.

Line 324 well if we use the SLIP as effective leg the shape is already given. Segments care only for minimization of joint work. Moreover, (Rode et al., 2016) showed that minimization of joint work explain kinematics in birds of different leg segments. Here elastic recoil paly a role but not for all leg segment configurations. If bipeds evolved form quadrupeds, one can expect that those principles are also valid for the forelimbs.

Line 340 I do not accept this explanation of why the authors use a two-segment model. One don't build a simple model just because one have relative low knowledge of how limbs work. A three-segment model is also not so complex as you just need to optimize one degree of freedom.

Lines 348-349 "there are often additional segments" what the authors mean? I do not know any animal leg with two segments.

Figure 4 Are the forces displayed simulated? Is the progression of the effective legs simulated? or both are some kind of mean from experiments? please explain

Results and explanation:

I have two problems with this section. The first one is again that a two-segment leg induce higher extension-flexion rates compared to a three or four segmented leg. So you are missing the real strategies that animals employ. Second, GRF in mammals have fore-aft forces. This means you are overestimating joint torque and power at beginning (hindlimb)/late(forelimb) stance.

General conclusions:

Line 410 Again it is not so new as already Rode and colleagues showed that more vertical oriented forces minimizing joint work predict leg segment kinematics. And they showed this with real animal data.

Line 416 Here the authors should explain the deviation of the model related to reality and not only mention it.

Aminiaghdam, S., Rode, C., Müller, R. and Blickhan, R. (2017). Increasing trunk flexion transforms human leg function into that of birds despite different leg morphology. *Journal of Experimental Biology* 220, 478-486.

Andrada, E., Rode, C., Sutedja, Y., Nyakatura, J. A. and Blickhan, R. (2014). Trunk orientation causes asymmetries in leg function in small bird terrestrial locomotion. *Proceedings of the Royal Society B: Biological Sciences* 281.

Blickhan, R., Andrada, E., Müller, R., Rode, C. and Ogihara, N. (2015). Positioning the hip with respect to the COM: Consequences for leg operation. *Journal of theoretical biology* 382, 187-197.

Geyer, H., Seyfarth, A. and Blickhan, R. (2006). Compliant leg behaviour explains basic dynamics of walking and running. *Proc. R. Soc. B* 273, 2861-7.

Gunther, M., Keppler, V., Seyfarth, A. and Blickhan, R. (2004). Human leg design: optimal axial alignment under constraints. *J Math Biol* 48, 623-46.

Maus, H. M., Lipfert, S. W., Gross, M., Rummel, J. and Seyfarth, A. (2010). Upright human gait did not provide a major mechanical challenge for our ancestors. *Nature communications* 1, 70.

Nyakatura, J. A., Melo, K., Horvat, T., Karakasiliotis, K., Allen, V. R., Andikfar, A., Andrada, E., Arnold, P., Lauströer, J., Hutchinson, J. R. et al. (2019). Reverse-engineering the locomotion of a stem amniote. *Nature* 565, 351-355.

Rode, C., Sutedja, Y., Kilbourne, B. M., Blickhan, R. and Andrada, E. (2016). Minimizing the cost of locomotion with inclined trunk predicts crouched leg kinematics of small birds at realistic levels of elastic recoil. *Journal of Experimental Biology* 219, 485-490.

Seyfarth, A., Günther, M. and Blickhan, R. (2001). Stable operation of an elastic three-segment leg. *Biological cybernetics* 84, 365-382.

Witte, H., Hackert, R., Fischer, M., Ilg, W., Albiez, J., Dillmann, R. and Seyfarth, A. (2001). Design criteria for the leg of a walking machine derived by biological inspiration from quadrupedal mammals. In *Proc. CLAWAR'2001-4th Int. Conf. on Climbing and Walking Robots*, pp. 63-68.

Review form: Reviewer 2

Recommendation

Accept with minor revision (please list in comments)

Scientific importance: Is the manuscript an original and important contribution to its field?

Good

General interest: Is the paper of sufficient general interest?

Good

Quality of the paper: Is the overall quality of the paper suitable?

Good

Is the length of the paper justified?

Yes

Should the paper be seen by a specialist statistical reviewer?

No

Do you have any concerns about statistical analyses in this paper? If so, please specify them explicitly in your report.

No

It is a condition of publication that authors make their supporting data, code and materials available - either as supplementary material or hosted in an external repository. Please rate, if applicable, the supporting data on the following criteria.

Is it accessible?

No

Is it clear?

No

Is it adequate?

No

Do you have any ethical concerns with this paper?

Yes

Comments to the Author

I have read RSPB-2020-1517. The authors use modeling, comparative ground-reaction force data, limb-geometric considerations, and phylogenetics mapping to devise some broadly applicable principles about quadrupedal locomotion across tetrapods.

I like this paper, and whereas I am often a nitpicky reviewer, there were few nits to pick. I think this paper is thought-provoking and you have done a good job with carefully outlining your assumptions and couching your interpretations as it comes to how and when the ideas apply, as well as limitations to causation etc. My comments below involve a couple of major points, several minor ones, and lastly some suggestions that might make this paper less of a challenging read.

Major:

Figure 4. I am confused by the first-to second position of the hind limb in A and the forelimb in B. These transitions would involve considerable moments at the knee (A) and elbow (B) respectively, yet the associated (changes in) velocity (\times force = power) are not seen in the power-curves for the respective conditions below. I don't think that this can be correct. from a modeling nor from a first-principles perspective. The same goes for the third-to-second last transition for the opposite limb joints in each condition (A and B). I doubt that animal joints operate in this herky-jerky fashion, but would of course be convinced by empirical data and I urge the authors to provide such evidence.

L 133. More information is needed about (1) the simulation environment, (2) the parameter settings, and (3) the underlying assumptions. Also, the numerical output from your simulations are not made available.

Table S1. To make it easier for the reader to locate your original data, I would encourage the authors to list source publications for these data, per row, and then use Granatosky et al., 2020 for the remainder.

Minor:

Abstract sentences one and two: There is quite a void between these two sentences. Cost is associated with generating force (i.e. expenditure of ATP). Work follows if force is associated with displacement. So, whilst there is movement economy associated with generating work, it is important that work is not the only, and certainly not the fundamental currency of movement-economy. Maybe delete the first sentence?

Line 59: Maybe provide taxonomical examples for the non-specialist; "such as birds and humans".

L. 66. The prior part of this sentence gives nothing about CoM elevation. And, even if it did, it would still be unclear what "low midstance" means. Maybe a diagram would help?

L. 91. I suggest you remove "perfect elastic mechanisms" they are not likely to exist.

L 106. "Appropriate passive linkages" I think this requires more explanation, or at least add some references to works describing such linkages more in-depth.

L 110. Maybe elaborate on this comment a bit; I think this is an important observation, as this practice is common in the human biomechanics literature.

L. 116. "Not included" in what? The author's deliberations, correct?

L. 119. Please cite some of the classic works that made this explicit. (e.g. Roberts et al., 1997, and others).

General problem (at least three places in the MS). Tetrapods come in both bi- and quadrupedal forms, so need to indicate which ones you are talking about.

Figures and tables:

L. 103. I think, for the uninitiated, it would be helpful to add more explanation for how sprawled stance can allow translation without work at joints. Maybe a diagram ala. Fig. 7 in Jenkins and Goslow 1983, J. Morph.

Fig. 1. Please add 0 to 1 labels to the y-axis.

Fig. 2. The rosette should have an axis indicating 0-100%. The entry with only hind limb information needs explanation. I would also acknowledge the basal lepidosaurian clade that defies the otherwise general trend you describe.

Style:

L. 66. Often using n-dash instead of comma's

L. 64. Example of references clustered at the end of complex sentence where many sub-statements require specific references.

L. 68. Broadly axial limb loading could be expressed clearer.

L. 77 Reword: step periods that in a relative sense are very long?

L. 87. Either "a walking alligator" or "walking alligators" - same for tortoise.

L. 176,389: moment arms can be great or large, not high.

Sentence starting L. 187. pretty intense, could you please break it up?

L. 190. I think the meaning of early-skewed would be clearer if more words were added to the sentence. Then, the meaning of "late skew" in L. 193 follows naturally.

L. 190. The meaning of "accounts" is not particularly clear here - I thought you were referring to previous studies as "accounts".

L. 195-6. Maybe clearer as "With a limb loading that is more vertical than axial"?

L. 342: replace "they" with "these costs".

L 354 with respect to the foot and located one leg length...4

L 384. reword: works to work contributions?

L386. replace the stance with either "the stance phase" or "stance"

No anonymity necessary: Nicolai Konow

Decision letter (RSPB-2020-1517.R0)

07-Sep-2020

Dear Jim:

Your manuscript has now been peer reviewed and the reviews have been assessed by an Associate Editor. The reviewers' comments (not including confidential comments to the Editor) and the comments from the Associate Editor are included at the end of this email for your reference. As you will see, the reviewers and the Editors have raised some concerns with your manuscript and we would like to invite you to revise your manuscript to address them.

Research ethics:

Use of animals and field studies:

It is a condition of publication that you make available the data and research materials supporting the results in the article. Please see our Data Sharing Policies (<https://royalsociety.org/journals/authors/author-guidelines/#data>). Datasets should be deposited in an appropriate publicly available repository and details of the associated accession number, link or DOI to the datasets must be included in the Data Accessibility section of the

article (<https://royalsociety.org/journals/ethics-policies/data-sharing-mining/>). Reference(s) to datasets should also be included in the reference list of the article with DOIs (where available).

Please submit a copy of your revised paper within three weeks. If we do not hear from you within this time your manuscript will be rejected. If you are unable to meet this deadline please let us know as soon as possible, as we may be able to grant a short extension.

Best wishes,
Sasha

Dr Sasha Dall
mailto: proceedingsb@royalsociety.org

Associate Editor
Comments to Author:

Dear Dr. Usherwood,

Thank you for submitting your manuscript entitled "A methodology to select mathematically valid fitting procedures to estimate the critical speed" to the Proceedings of the Royal Society. I have received two peer reviews, both of which find that the topic of the manuscript is of scientific interest to the readers of Proceedings B and are overall supportive of the manuscript.

I appreciate that your manuscript develops a functional explanation for the elbow-back knee-forward arrangement of leg joints in quadrupeds; the study develops predictions from mathematical models to compare the costs of two joint arrangements and compares the predictions with experimental observations (force measurements) from the literature. The main conclusion of the study is that the elbow-back knee-forward joint arrangement reduces the

mechanical work required for limbed locomotion more than an elbow-back knee-forward arrangement.

Both reviewers appreciate that this manuscript provides a valuable review and introduction into the topic of limb segmentation. The reviewers, however, also raise some concerns about the scope of the study and the clarity of the presentation. Reviewer 2 found the manuscript quite difficult to read at some points and has many suggestions to improve clarity and help the paper succeed as an introduction to this topic. Both reviewers recommend that the author more explicitly address the limitations of this study, such as limitations imposed by the assumptions of the model (simplifications such as the number of leg segments). Reviewer 1 in particular points out that there is a large body of literature, often on bipeds, that is highly relevant for this study.

There might be mechanisms other than collision mechanics at play that may lead to alternative hypotheses about leg segmentation that are currently insufficiently explored in this study.

Reviewer(s)' Comments to Author:

Referee: 1

Comments to the Author(s)

The authors build a nice history around a very simplified model and blend out important literature that already discussed about more vertical forces from trunk balance point of view e.g. (Aminiaghdam et al., 2017; Andrada et al., 2014; Blickhan et al., 2015; Maus et al., 2010). There exist also more literature about leg segmentation (Gunther et al., 2004; Seyfarth et al., 2001; Witte et al., 2001). Nyakatura and colleagues also used joint power to reconstruct walking gaits of salamanders, and of the extinct *Orabates* (Nyakatura et al., 2019). Knowing that literature, the results are so novel as the authors claim. Still the authors have a point, and if they go out of their very enclosed point of view, the manuscript can be an interesting lecture for biologist that start studying leg design.

Line 59; The authors start the sentence with „in bipeds“, however, they talk only about humans. The explanation that follows about pitch moment of inertia is not valid for birds. The authors know for sure that most bipeds are birds. The authors avoid here a lot of literature that already noted the balance of trunk necessitates more vertical ground reaction forces.

Line 84 This is just valid if you have a wheel or the motion is quasi-static. I think the authors should be fair here and state this. No mammal uses the strategy proposed by the authors, as all of them display fore-aft forces and contact time are not infinitesimal.

Section: Axial forces reduce joint work not limb work

I think the model is not clear enough described by the authors. Is the model dimensionless? If not what are the contact times?

I guess the motion of the two segments is constrained by the contraction/extension of the effective leg? If so, explain where the motion of the effective leg came from? From a rat?

Why the authors use a two-segment model? Of course, I understand that the kinematic of the segments is given (but not for every reader), but to me every simplification of reality must be explained and contextualized. For me is not enough to say we know there are others that used 3 segments, but we use 2. In addition

If you are going to build a robot, I would see your point, but here you are trying to explain form and function. Well, if you have two segments, the kinematic of the segments is different to the real segment kinematic. Thus, I need that you show that the power deviations relative to a 3/4 segment leg do not alters your point.

Lines 174/180: This explanation is completely sound for the simplified model proposed by the authors. However, it is not for a leg with more than 2 segments. If you have three segments you can reduce joint excursions and therefore reduce joint work. This is the way that most animals do. For example, dogs joint elbow/knee angles do not or vary little during stance. If you have two segments, joint angles will always display important excursions and therefore higher joint works.

Line 183 Well I do not agree, there are important literature showing that more vertical GRF are needed for trunk balance e.g. (Andrada et al., 2014; Blickhan et al., 2015; Maus et al., 2010). And that the skewed GRF to is related to the shift of the COM with respect to the pelvis (Aminiaghdam et al., 2017; Andrada et al., 2014).

Line 194 this is because the most proximal joint is located cranially to the COM. Following Aminiaghdam et al. 2017 a person walking prone backwards should display the late-skew. About bipedy and skewed GRFs. The bipedal SLIP model produces both asymmetries. Left-skewed is self-stable right-skewed is not.

Line 318 There is another explanation for the M-shape of the GRF. While the inverted pendulum and collision models cannot reproduce the shape of the GRF, the SLIP model does (Geyer et al., 2006). Again, the authors seem to blend out literature that do not agree with their claims.

Line 324 well if we use the SLIP as effective leg the shape is already given. Segments care only for minimization of joint work. Moreover, (Rode et al., 2016) showed that minimization of joint work explain kinematics in birds of different leg segments. Here elastic recoil play a role but not for all leg segment configurations. If bipeds evolved from quadrupeds, one can expect that those principles are also valid for the forelimbs.

Line 340 I do not accept this explanation of why the authors use a two-segment model. One don't build a simple model just because one have relative low knowledge of how limbs work. A three-segment model is also not so complex as you just need to optimize one degree of freedom.

Lines 348-349 "there are often additional segments" what the authors mean? I do not know any animal leg with two segments.

Figure 4 Are the forces displayed simulated? Is the progression of the effective legs simulated? or both are some kind of mean from experiments? please explain

Results and explanation:

I have two problems with this section. The first one is again that a two-segment leg induce higher extension-flexion rates compared to a three or four segmented leg. So you are missing the real strategies that animals employ. Second, GRF in mammals have fore-aft forces. This means you are overestimating joint torque and power at beginning (hindlimb)/late (forelimb) stance.

General conclusions:

Line 410 Again it is not so new as already Rode and colleagues showed that more vertical oriented forces minimizing joint work predict leg segment kinematics. And they showed this with real animal data.

Line 416 Here the authors should explain the deviation of the model related to reality and not only mention it.

Aminiaghdam, S., Rode, C., Müller, R. and Blickhan, R. (2017). Increasing trunk flexion transforms human leg function into that of birds despite different leg morphology. *Journal of Experimental Biology* 220, 478-486.

Andrada, E., Rode, C., Suttedja, Y., Nyakatura, J. A. and Blickhan, R. (2014). Trunk orientation causes asymmetries in leg function in small bird terrestrial locomotion. *Proceedings of the Royal Society B: Biological Sciences* 281.

Blickhan, R., Andrada, E., Müller, R., Rode, C. and Ogihara, N. (2015). Positioning the hip with respect to the COM: Consequences for leg operation. *Journal of theoretical biology* 382, 187-197.

Geyer, H., Seyfarth, A. and Blickhan, R. (2006). Compliant leg behaviour explains basic dynamics of walking and running. *Proc. R. Soc. B* 273, 2861-7.

Gunther, M., Keppler, V., Seyfarth, A. and Blickhan, R. (2004). Human leg design: optimal axial alignment under constraints. *J Math Biol* 48, 623-46.

Maus, H. M., Lipfert, S. W., Gross, M., Rummel, J. and Seyfarth, A. (2010). Upright human gait did not provide a major mechanical challenge for our ancestors. *Nature communications* 1, 70.

- Nyakatura, J. A., Melo, K., Horvat, T., Karakasiliotis, K., Allen, V. R., Andikfar, A., Andrada, E., Arnold, P., Lauströer, J., Hutchinson, J. R. et al. (2019). Reverse-engineering the locomotion of a stem amniote. *Nature* 565, 351-355.
- Rode, C., Sutedja, Y., Kilbourne, B. M., Blickhan, R. and Andrada, E. (2016). Minimizing the cost of locomotion with inclined trunk predicts crouched leg kinematics of small birds at realistic levels of elastic recoil. *Journal of Experimental Biology* 219, 485-490.
- Seyfarth, A., Günther, M. and Blickhan, R. (2001). Stable operation of an elastic three-segment leg. *Biological cybernetics* 84, 365-382.
- Witte, H., Hackert, R., Fischer, M., Ilg, W., Albiez, J., Dillmann, R. and Seyfarth, A. (2001). Design criteria for the leg of a walking machine derived by biological inspiration from quadrupedal mammals. In *Proc. CLAWAR'2001-4th Int. Conf. on Climbing and Walking Robots*, pp. 63-68.

Referee: 2

Comments to the Author(s)

I have read RSPB-2020-1517. The authors use modeling, comparative ground-reaction force data, limb-geometric considerations, and phylogenetics mapping to devise some broadly applicable principles about quadrupedal locomotion across tetrapods.

I like this paper, and whereas I am often a nitpicky reviewer, there were few nits to pick. I think this paper is thought-provoking and you have done a good job with carefully outlining your assumptions and couching your interpretations as it comes to how and when the ideas apply, as well as limitations to causation etc. My comments below involve a couple of major points, several minor ones, and lastly some suggestions that might make this paper less of a challenging read.

Major:

Figure 4. I am confused by the first-to second position of the hind limb in A and the forelimb in B. These transitions would involve considerable moments at the knee (A) and elbow (B) respectively, yet the associated (changes in) velocity (\times force = power) are not seen in the power-curves for the respective conditions below. I don't think that this can be correct. from a modeling nor from a first-principles perspective. The same goes for the third-to-second last transition for the opposite limb joints in each condition (A and B). I doubt that animal joints operate in this herky-jerky fashion, but would of course be convinced by empirical data and I urge the authors to provide such evidence.

L 133. More information is needed about (1) the simulation environment, (2) the parameter settings, and (3) the underlying assumptions. Also, the numerical output from your simulations are not made available.

Table S1. To make it easier for the reader to locate your original data, I would encourage the authors to list source publications for these data, per row, and then use Granatosky et al., 2020 for the remainder.

Minor:

Abstract sentences one and two: There is quite a void between these two sentences. Cost is associated with generating force (i.e. expenditure of ATP). Work follows if force is associated with displacement. So, whilst there is movement economy associated with generating work, it is important that work is not the only, and certainly not the fundamental currency of movement-economy. Maybe delete the first sentence?

Line 59: Maybe provide taxonomical examples for the non-specialist; "such as birds and humans".

L. 66. The prior part of this sentence gives nothing about CoM elevation. And, even if it did, it would still be unclear what "low midstance" means. Maybe a diagram would help?

L. 91. I suggest you remove "perfect elastic mechanisms" they are not likely to exist.

L 106. "Appropriate passive linages" I think this requires more explanation, or at least add some references to works describing such linkages more in-depth.

L 110. Maybe elaborate on this comment a bit; I think this is an important observation, as this practice is common in the human biomechanics literature.

L. 116. "Not included" in what? The author's deliberations, correct?

L. 119. Please cite some of the classic works that made this explicit. (e.g. Roberts et al., 1997, and others).

General problem (at least three places in the MS). Tetrapods come in both bi- and quadrupedal forms, so need to indicate which ones you are talking about.

Figures and tables:

L. 103. I think, for the uninitiated, it would be helpful to add more explanation for how sprawled stance can allow translation without work at joints. Maybe a diagram ala. Fig. 7 in Jenkins and Goslow 1983, J. Morph.

Fig. 1. Please add 0 to 1 labels to the y-axis.

Fig. 2. The rosette should have an axis indicating 0-100%. The entry with only hind limb information needs explanation. I would also acknowledge the basal lepidosaurian clade that defies the otherwise general trend you describe.

Style:

L. 66. Often using n-dash instead of comma's

L. 64. Example of references clustered at the end of complex sentence where many sub-statements require specific references.

L. 68. Broadly axial limb loading could be expressed clearer.

L. 77 Reword: step periods that in a relative sense are very long?

L. 87. Either "a walking alligator" or "walking alligators" - same for tortoise.

L. 176,389: moment arms can be great or large, not high.

Sentence starting L. 187. pretty intense, could you please break it up?

L. 190. I think the meaning of early-skewed would be clearer if more words were added to the sentence. Then, the meaning of "late skew" in L. 193 follows naturally.

L. 190. The meaning of "accounts" is not particularly clear here - I thought you were referring to previous studies as "accounts".

L. 195-6. Maybe clearer as "With a limb loading that is more vertical than axial"?

L. 342: replace "they" with "these costs".

L 354 with respect to the foot and located one leg length...4

L 384. reword: works to work contributions?

L386. replace the stance with either "the stance phase" or "stance"

No anonymity necessary: Nicolai Konow

Author's Response to Decision Letter for (RSPB-2020-1517.R0)

See Appendix A.

RSPB-2020-1517.R1 (Revision)

Review form: Reviewer 1

Recommendation

Accept as is

Scientific importance: Is the manuscript an original and important contribution to its field?

Good

General interest: Is the paper of sufficient general interest?

Good

Quality of the paper: Is the overall quality of the paper suitable?

Excellent

Is the length of the paper justified?

Yes

Should the paper be seen by a specialist statistical reviewer?

No

Do you have any concerns about statistical analyses in this paper? If so, please specify them explicitly in your report.

No

It is a condition of publication that authors make their supporting data, code and materials available - either as supplementary material or hosted in an external repository. Please rate, if applicable, the supporting data on the following criteria.

Is it accessible?

Yes

Is it clear?

Yes

Is it adequate?

Yes

Do you have any ethical concerns with this paper?

No

Comments to the Author

Looking forward to see your MS published!

Review form: Reviewer 2**Recommendation**

Accept as is

Scientific importance: Is the manuscript an original and important contribution to its field?

Good

General interest: Is the paper of sufficient general interest?

Good

Quality of the paper: Is the overall quality of the paper suitable?

Good

Is the length of the paper justified?

Yes

Should the paper be seen by a specialist statistical reviewer?

No

Do you have any concerns about statistical analyses in this paper? If so, please specify them explicitly in your report.

No

It is a condition of publication that authors make their supporting data, code and materials available - either as supplementary material or hosted in an external repository. Please rate, if applicable, the supporting data on the following criteria.

Is it accessible?

No

Is it clear?

N/A

Is it adequate?

N/A

Do you have any ethical concerns with this paper?

No

Comments to the Author

I thank the authors for their revision efforts, which have addressed my comments and suggestions to a satisfactory extent.

Decision letter (RSPB-2020-1517.R1)

23-Oct-2020

Dear Jim

I am pleased to inform you that your manuscript RSPB-2020-1517.R1 entitled "Limb work and joint work minimisation reveal an energetic benefit to the elbows-back, knees-forward limb design in parasagittal quadrupeds" has been accepted for publication in Proceedings B.

The referee(s) have recommended publication, but also suggest some minor revisions to your manuscript. Therefore, I invite you to respond to the referee(s)' comments and revise your manuscript. Because the schedule for publication is very tight, it is a condition of publication that you submit the revised version of your manuscript within 7 days. If you do not think you will be able to meet this date please let us know.

Sincerely,
Sasha

Dr Sasha Dall
Editor, Proceedings B
<mailto:proceedingsb@royalsociety.org>

Associate Editor:
Board Member: 1
Comments to Author:

Dear Dr. Usherwood,

although the manuscript states that the simulation outputs are available in the ESM, the data were not accessible to the reviewers and appear not to have been submitted as supplementary files. Please note that Proceedings B encourages authors to make their data available for peer review and, to the best of my knowledge, requires that authors make their data available upon publication. Please ensure that you submit all supplementary materials.

Reviewer(s)' Comments to Author:

Referee: 2

Comments to the Author(s)

I thank the authors for their revision efforts, which have addressed my comments and suggestions to a satisfactory extent.

Referee: 1

Comments to the Author(s)

Looking forward to see your MS published!

Author's Response to Decision Letter for (RSPB-2020-1517.R1)

See Appendix B.

Decision letter (RSPB-2020-1517.R2)

13-Nov-2020

Dear Dr Usherwood

I am pleased to inform you that your manuscript entitled "Limb work and joint work minimisation reveal an energetic benefit to the elbows-back, knees-forward limb design in parasagittal quadrupeds" has been accepted for publication in Proceedings B.

Your article has been estimated as being 9 pages long. Our Production Office will be able to confirm the exact length at proof stage.

Open Access

Paper charges

All supplementary materials accompanying an accepted article will be treated as in their final form. They will be published alongside the paper on the journal website and posted on the online

figshare repository. Files on figshare will be made available approximately one week before the accompanying article so that the supplementary material can be attributed a unique DOI.

Sincerely,
Editor, Proceedings B
<mailto:proceedingsb@royalsociety.org>

Appendix A

Responses to referees

We are most grateful for the considerable expert attention and investment in improving this manuscript. Accordingly, we have gone through the manuscript and have done our best to incorporate the reviewer's suggestions. We believe that these additions/revisions have greatly improved the quality of the work. We hope the revised manuscript is suitable for publication.

Referee: 1

I (Usherwood) must first apologise to referee 1 if I have caused offense. There is always a challenge in such a paper in determining which previous work is most relevant. My survey of the bipedal literature was certainly incomplete, largely because I viewed it as difficult to relate immediately to the quadrupedal case. It may be that this difficulty is interesting and informative: we now expand on this point and adopt the majority of indicated references.

I am particularly grateful for the highlighting of the Rode et al. paper – I had missed this and will certainly seek to add it to my Z-limb paper currently under review.

Comments to the Author(s)

The authors build a nice history around a very simplified model and blend out important literature that already discussed about more vertical forces from trunk balance point of view e.g. (Aminiaghdam et al., 2017; Andrada et al., 2014; Blickhan et al., 2015; Maus et al., 2010).

There exist also more literature about leg segmentation (Gunther et al., 2004; Seyfarth et al., 2001; Witte et al., 2001).

Nyakatura and colleagues also used joint power to reconstruct walking gaits of salamanders, and of the extinct Orabates (Nyakatura et al., 2019). Knowing that literature, the results are so novel as the authors claim. Still the authors have a point, and if they go out of their very enclosed point of view, the manuscript can be an interesting lecture for biologist that start studying leg design.

General comment: the majority of these references are now incorporated.

Line 59; The authors start the sentence with „in bipeds“, however, they talk only about humans. The explanation that follows about pitch moment of inertia is not valid for birds. The authors know for sure that most bipeds are birds. The authors avoid here a lot of literature that already noted the balance of trunk necessitates more vertical ground reaction forces.

We appreciate this point and remove the comment about collocated hips. However, we maintain that this passage applies to bipeds generally (humans, birds, rodent and marsupial hoppers). At this stage we are not claiming that forces are actually aligned axially, but that non-axial forces might produce pitching motions with energetic consequences.

Line 84 This is just valid if you have a wheel or the motion is quasi-static. I think the

authors should be fair here and state this. No mammal uses the strategy proposed by the authors, as all of them display fore-aft forces and contact time are not infinitesimal.

We completely agree with the review and is indeed partly the point being made, expanded on later. To signpost this issue, we now add:

'Force profiles of walking alligator and tortoise appear highly consistent with this wheel-like or 'sliding' strategy ([Usherwood, 2020] (this is not typical for mammals, particularly the familiar, larger species – see below)'

Section: Axial forces reduce joint work not limb work

I think the model is not clear enough described by the authors. Is the model dimensionless? If not what are the contact times?

We thank the reviewer for this suggestion. Model clarity is of the utmost importance. Accordingly, this section is now expanded in the text, and in electronic supplementary information:

The model is intended to be generic, so absolute lengths, speeds and timings are unimportant (powers are presented in Arbitrary Units); however, the geometry is broadly appropriate for a human (1m leg length) sprinting at 8m/s with both stance and aerial duration of 0.125s, landing with a slightly flexed leg (initial start height of 0.85m).

I guess the motion of the two segments is constrained by the contraction/extension of the effective leg? If so, explain where the motion of the effective leg came from? From a rat?

Following the reviewer's suggestion, we have expanded description of the model. This is now explicit:

Leg flexion/extension is prescribed by the centre of mass trajectory and the feasible geometry of the 2-segment leg.

Why the authors use a two-segment model? Of course, I understand that the kinematic of the segments is given (but not for every reader), but to me every simplification of reality must be explained and contextualized. For me is not enough to say we know there are others that used 3 segments, but we use 2. In addition

If you are going to build a robot, I would see your point, but here you are trying to explain form and function. Well, if you have two segments, the kinematic of the segments is different to the real segment kinematic. Thus, I need that you show that the power deviations relative to a 3/4 segment leg do not alters your point.

Extending to the 3-segment leg is indeed valuable, but is certainly beyond the scope of this study. Work minimization with 3-link models have begun to be pursued, but the manifold of configurations tend to mean that optimization approaches are required, making the geometric fundamental principles somewhat difficult to perceive. With a skewed force profile, work minimization through computer optimization does indeed arrive at the knees-forward, ankles-backward configuration in birds (Rode et al., 2016) and wallabies (Usherwood and McGowan, submitted). At this stage, all we can do is acknowledge that it is an assumption that a 2-segment leg is an informative reduction of an N-link leg; though we highlight this has not been demonstrated. In the

caveats section:

The caveats concerning deviation from reality (especially neglecting the action of the shoulder in the forelimb or ankle-toe segment of the hindlimb) should be remembered. The modelling and interpretation presented here relies on a 2-segment reduction of quadruped legs; it remains to be determined whether the same principles apply to multi-segment limbs (bear in mind that horse legs might be considered as consisting of seven segments, with seven joints!).

Lines 174/180: This explanation is completely sound for the simplified model proposed by the authors. However, it is not for a leg with more than 2 segments. If you have three segments you can reduce joint excursions and therefore reduce joint work. This is the way that most animals do. For example, dogs joint elbow/knee angles do not or vary little during stance. If you have two segments, joint angles will always display important excursions and therefore higher joint works.

We would agree that 3 (or perhaps more) segments would have the potential to reduce joint work further – we do not claim that the 2-segment leg is optimal. However, the principle by which joint work can be reduced might best be described beginning with the simplest, 2-segment case.

Line 183 Well I do not agree, there are important literature showing that more vertical GRF are needed for trunk balance e.g. (Andrada et al., 2014; Blickhan et al., 2015; Maus et al., 2010). And that the skewed GRF to is related to the shift of the COM with respect to the pelvis (Aminiaghdam et al., 2017; Andrada et al., 2014). Line 194 this is because the most proximal joint is located cranially to the COM. Following Aminiaghdam et al. 2017 a person walking prone backwards should display the late-skew. About bipedy and skewed GRFs. The bipedal SLIP model produces both asymmetries. Left-skewed is self-stable right-skewed is not.

We agree with the reviewer that stability is indeed an interesting area, and has been used as accounts for various features of force profiles and orientations in bipeds. However, it does not appear a suitable account for quadrupeds. We expand on this point and give due acknowledgement to the previous bipedal work, but also to highlight the contrast with the quadrupedal case, and perhaps the value in returning to issues of work. We hope this provides the reader with the appropriate background information to make informed assessments of our model.

Forces directed more vertically than axially indicate a pressure towards limb work rather than joint work minimisation. Were this to be observed, it would suggest – but not explicitly demonstrate – at least some degree of inter-joint power transfer. Measured vertical force profiles provide evidence of more-vertical-than-axial limb loading. It has long been recognised ([Alexander and Vernon, 1975]) that vertical forces would demand lower work requirements than axial, but also identified that this is not consistent with observed force orientation in walking and running animals. This was attributed to an absence – or at least insufficiency – of multi-joint linkages to provide inter-joint power transfer ([Alexander and Vernon, 1975]), meaning that minimisation of joint work was taken as a suitable initial cost function to consider when exploring animal limb design (though see also [Kuznetsov, 1995]). Minimisation of joint work does appear effective in accounting for some features of bipedal ([Rode et al., 2016]) and quadrupedal ([Polet and Bertram, 2019]) gait kinetics ([Alexander, 1980]). But force vectors in bipeds (humans: e.g. [Maus et al., 2010]; birds: [Andrada et al., 2014; wallabies;] McGowan et al., 2005) and quadrupeds ([Ayres and Alexander, 1978]) are consistently observed to be orientated between axial and vertical. If joint work was indeed the suitable energetic cost, some non-energetic explanation is then required. Stability is one such consideration ([Maus et al., 2010]), and has driven considerable interest, not least due to the very demanding challenges

of achieving stability in bipedal robots. However, the stability issue does not appear to translate to quadrupeds: with a suitable pitch moment of inertia, stability in pitch is easily achieved ([Murphy, 1984]). In quadrupeds at least, then, we interpret orientation of force vectors between axial and vertical as indicating a compromise between joint work (axial) and limb work (vertical) minimisation.

Vertical forces that are higher in early stance than late may be termed early-skewed. These are widely observed in bipeds, including young children, human sprinters and birds. Potential explanations for this range from energetic (Children: [Usherwood et al., 2018; birds: [Birn-Jeffery et al., 2014]) to stability ([Geyer et al., 2006]) to anatomical (sprinters: [Clark et al., 2014; birds: [Andrada et al., 2014; [Bishop et al., 2018]). Whatever account for early skew is favoured for bipeds, late skew – a common feature of quadruped forelimb vertical force profiles (Fig. 2) – is not consistent with any of the accounts for skew in bipeds. Instead, it points to pitch avoidance with a limb loading that is more vertical than purely axial ([Jayes and Alexander, 1980]), a feature of low limb-work gaits ([Usherwood, 2020]). Vertical forces from the forelimb in early stance act on a large moment arm about the centre of mass, which reduces over the duration of stance; in order to impose both a net-zero pitching moment and constant weight support, this requires a vertical force profile that increases through stance, accounting for the late-skew.

Line 318 There is another explanation for the M-shape of the GRF. While the inverted pendulum and collision models cannot reproduce the shape of the GRF, the SLIP model does (Geyer et al., 2006). Again, the authors seem to blend out literature that do not agree with their claims.

Again, apologies for offense caused. We now include a reference to the Geyer study, though explicitly continue the ‘story’ from a work minimizing (and so vaulting) rather than spring-dominated perspective. An explicit review of the inverted pendulum and collision models versus the SLIP model is beyond the scope of this study.

Line 324 well if we use the SLIP as effective leg the shape is already given. Segments care only for minimization of joint work. Moreover, (Rode et al., 2016) showed that minimization of joint work explain kinematics in birds of different leg segments. Here elastic recoil paly a role but not for all leg segment configurations. If bipeds evolved form quadrupeds, one can expect that those principles are also valid for the forelimbs.

We understand the reviewer’s point on this issue, but believe such an expansion of the SLIP model is beyond the focus of this work and distracts the reader. We further highlight that the development focuses on energetic issues rather than spring-mass models:

Among energetic (as opposed to spring-dominated) accounts...

Line 340 I do not accept this explanation of why the authors use a two-segment model. One don’t build a simple model just because one have relative low knowledge of how limbs work. A three-segment model is also not so complex as you just need to optimize one degree of freedom.

Simplified models are valuable not only due to ease of computation, but also because they can be more tractable in terms of mechanism. As such, the 2-segment model appears suitable, bridging the familiar single-segment spring-mass (SLIP) model and the 3-segment models that result in multi-dimensional cost parameter spaces that are difficult to interpret. We have provided the reader with potential limitations of the study in the caveats section of the manuscript.

Lines 348-349 “there are often additional segments” what the authors mean? I do not know any animal leg with two segments.

To provide additional clarification for the reader to acknowledge the limitations of a two-segment model, we have reworded the text.

In reality, depending on level of biological realism demanded, there may be one ([Kuznetsov1995, Fischer and Blickhan, 2006]) or multiple additional segments,,

Figure 4 Are the forces displayed simulated? Is the progression of the effective legs simulated? or both are some kind of mean from experiments? please explain

The kinematics and forces are entirely modeled – now further highlighted:

This allows the joint power profiles to be determined (Fig. 4) given modelled 2-segment limb kinematics and theoretical force profiles

Results and explanation:

I have two problems with this section. The first one is again that a two-segment leg induce higher extension-flexion rates compared to a three or four segmented leg. So you are missing the real strategies that animals employ.

At some level of detail, this is clearly true. We do, however, see a value in this intermediate reduction – somewhere between viewing a leg as a linear actuator and attempting to include all the segments and connections – in providing insight at a tractable, mechanistic level.

Second, GRF in mammals have fore-aft forces. This means you are overestimating joint torque and power at beginning (hindlimb)/late(forelimb) stance.

We acknowledge that mammals have fore-aft forces. This manuscript is presenting a ‘what if’ model, pushing the reductions to the extreme in order to enable explicit discussion of mechanism.

General conclusions:

Line 410 Again it is not so new as already Rode and colleagues showed that more vertical oriented forces minimizing joint work predict leg segment kinematics. And they showed this with real animal data.

Indeed, this is an important paper that is now cited (and I shall seek to cite it when revising my current Z-limb submission). We thank the reviewer for providing this reference.

Line 416 Here the authors should explain the deviation of the model related to reality and not only mention it.

The caveats section is duly expanded:

The modelling and interpretation presented here relies on a 2-segment reduction of quadruped legs; it remains to be determined whether the same principles apply to multi-segment limbs (bear in mind that horse legs might be considered as consisting of seven segments, with seven joints!). Also, the current model is parasagittal and planar; no real animal limb operates in precisely such a manner. Further, alternative hypotheses relating to the implications of elbow/knee orientation concerning directional stability ([Lee and Meek, 2005] or passive release of elastic energy facilitating protraction ([Wilson et al., 2003]) should not be dismissed, especially for highly cursorial quadrupeds. And multi-segment limbs may well offer multiple advantages, from facilitating elastic storage and recoil (see [Rode et al., 2016]) with implications in terms of passive joint stabilisation ([Seyfarth et al., 2001]).

Referee: 2

Comments to the Author(s)

I have read RSPB-2020-1517. The authors use modeling, comparative ground-reaction force data, limb-geometric considerations, and phylogenetics mapping to devise some broadly applicable principles about quadrupedal locomotion across tetrapods.

I like this paper, and whereas I am often a nitpicky reviewer, there were few nits to pick. I think this paper is thought-provoking and you have done a good job with carefully outlining your assumptions and couching your interpretations as it comes to how and when the ideas apply, as well as limitations to causation etc. My comments below involve a couple of major points, several minor ones, and lastly some suggestions that might make this paper less of a challenging read.

Major:

Figure 4. I am confused by the first-to second position of the hind limb in A and the forelimb in B. These transitions would involve considerable moments at the knee (A) and elbow (B) respectively, yet the associated (changes in) velocity (x force = power) are not seen in the power-curves for the respective conditions below. I don't think that this can be correct. from a modeling nor from a first-principles perspective.

After reviewing the figure, we respectfully disagree with the reviewer (i.e., resulting powers do look correct from a first-principles perspective). In both cases the initial moment arms for the hip and knee are high (the foot is fully extended forward) as are the angular velocities (due to this initial rapid flexion). In the hind limb of A there are high forces at this instant, resulting in a high initial joint power – as described above and visible in the power plot. However, this rapidly diminishes as the distal segment soon becomes vertical, resulting in low moment arm for the 'knee', and foot motion is due to rotation of this vertical segment, resulting in low angular velocity of the hip. So, the joint power begins high but falls past zero by 20% stance. In the forelimb of B, the initial high power is not observed because the forces begin very low.

The same goes for the third-to-second last transition for the opposite limb joints in each condition (A and B). I doubt that animal joints operate in this herky-jerky fashion, but would of course be convinced by empirical data and I urge the authors to provide such evidence.

Some parts of the modelled stance do indeed appear herky-jerky and biologically unrealistic. However, it does allow the geometry of the complete potential range of stance for a 2-segment leg to be described and modeled. The findings can be seen not to be down to the peculiarity of these straight-to-first-flexed instant or last-flexed-to-straight instants: high joint power demands continue through the first half of stance.

L 133. More information is needed about (1) the simulation environment, (2) the parameter settings, and (3) the underlying assumptions.

Following the reviewer's suggestion, we have expanded on points 1,2, and 3 in supplementary information, and code is provided.

Also, the numerical output from your simulations are not made available.

Simulation output resulting in figures is now provided in ESM

Table S1. To make it easier for the reader to locate your original data, I would encourage the authors to list source publications for these data, per row, and then use Granatosky et al., 2020 for the remainder.

Table adjusted

Minor:

Abstract sentences one and two: There is quite a void between these two sentences. Cost is associated with generating force (i.e. expenditure of ATP). Work follows if force is associated with displacement. So, whilst there is movement economy associated with generating work, it is important that work is not the only, and certainly not the fundamental currency of movement-economy. Maybe delete the first sentence?

Difficult. Both sentences are true, and both describe the motivation of the study. It is not claimed that the physiological costs are down 'only' to mechanical work, but the assumption is that some form of mechanical work has 'relevance' to energetic cost. That metabolic cost correlates well with 'generating force' is true, but may also be a misleading summary. Trivial modifications can be imagined in terms of both limb forces (higher duty factor) or muscle loading (straighter limbs) that would reduce the force and – if truly causal – therefore cost. However, this is does not work. Locomotion up an incline certainly demands more ATP cost. The view that it is not a mechanical work or power that acts as the fundamental driver of metabolic cost is largely due to the poor correlation between any measured forms of mechanical work and oxygen consumption. However, it is unclear what form of mechanical work might be most relevant – and this is partly why exploring different forms of mechanical work continues to be important.

Line 59: Maybe provide taxonomical examples for the non-specialist; "such as birds and humans".

Added:

In bipeds (humans, birds, rodent and marsupial hoppers)

L. 66. The prior part of this sentence gives nothing about CoM elevation. And, even if it did, it would still be unclear what "low midstance" means. Maybe a diagram would help?

Passage extended for clarity:

In running, axially loaded limbs slow the body both horizontally and vertically until the centre of mass is relatively low and the supporting leg flexed, requiring work to re-accelerate and lift the body during the second half of stance.

L. 91. I suggest you remove “perfect elastic mechanisms” they are not likely to exist.

This is intended as a thought experiment to demonstrate extremes. This is now explicit:

For instance, with hypothetical perfect elastic mechanisms, or suitable transfer between centre of mass and rotational energies

L 106. "Appropriate passive linkages" I think this requires more explanation, or at least add some references to works describing such linkages more in-depth.

We have expanded the text and provide the reader with an appropriate citation.

in contrast, parasagittal limbs with predominantly transverse-axis joints must experience simultaneous and cancelling positive and negative joint powers, but this could be achieved with appropriate passive linkages ([Usherwood, 2020]).

L 110. Maybe elaborate on this comment a bit; I think this is an important observation, as this practice is common in the human biomechanics literature.

Hopefully the expanded section covering some of the history leading to the focus on joint work will help here.

It has long been recognised ([Alexander and Vernon, 1975]) that vertical forces would demand lower work requirements than axial, but also identified that this is not consistent with observed force orientation in walking and running animals. This was attributed to an absence – or at least insufficiency – of multi-joint linkages to provide inter-joint power transfer ([Alexander and Vernon, 1975]), meaning that minimisation of joint work was taken as a suitable initial cost function to consider when exploring animal limb design (though see also [Kuznetsov, 1995]). Minimisation of joint work does appear effective in accounting for some features of bipedal ([Rode et al., 2016]) and quadrupedal ([Polet and Bertram, 2019]) gait kinetics ([Alexander, 1980]). But force vectors in bipeds (humans: e.g. [Maus et al., 2010; birds: [Andrada et al., 2014; wallabies; [McGowan et al., 2005]) and quadrupeds ([Jays and Alexander, 1978]) are consistently observed to be orientated between axial and vertical. If joint work was indeed the suitable energetic cost, some non-energetic explanation is then required. Stability is one such consideration ([Maus et al., 2010]), and has driven considerable interest, not least due to the very demanding challenges of achieving stability in bipedal robots. However, the stability issue does not appear to translate to quadrupeds: with a suitable pitch moment of inertia, stability in pitch is easily achieved ([Murphy, 1984]). In quadrupeds at least, then, we interpret orientation of force vectors between axial and vertical as indicating a compromise between joint work (axial) and limb work (vertical) minimisation.

L. 116. "Not included" in what? The author's deliberations, correct?

True. Rewording indeed required:

work and power associated with accelerating limb masses for protraction and retraction may be significant; as may be the costs associated with activating muscle isometrically

L. 119. Please cite some of the classic works that made this explicit. (e.g. Roberts et al., 1997, and others).

The point here is does not really concern tendon compliance as a means to reduce muscle work – this is now made more explicit (and a reference to Roberts et al., 1997 duly made).

However, weight support during travel with low mechanical work demand on muscle – whatever the other sources of physiological cost – is presumably an important feature of economical locomotion, and traits that facilitate this (in addition to tendon elastic energy storage and recoil, e.g. [Roberts et al., 1997] may provide insight into animal form and function.

General problem (at least three places in the MS). Tetrapods come in both bi- and quadrupedal forms, so need to indicate which ones you are talking about.

Expanded to 'quadrupedal tetrapods' where appropriate

Figures and tables:

L. 103. I think, for the uninitiated, it would be helpful to add more explanation for how sprawled stance can allow translation without work at joints. Maybe a diagram ala. Fig. 7 in Jenkins and Goslow 1983, J. Morph.

Again, this is most explicitly explored in Usherwood, 2020. I could cite this again (though it is on the previous line). I understand that some of the linkage concepts may not be familiar to some readers. The concepts of Usherwood, 2020 are summarized, but a more expansive development and justification would be difficult to fit within the current paper.

Fig. 1. Please add 0 to 1 labels to the y-axis.

Putting units on the y axis would appear arbitrary: they are Arbitrary Units, so the axis could range from 0 to 10000 as much as 0 to 1.

Fig. 2. The rosette should have an axis indicating 0-100%. The entry with only hind limb information needs explanation. I would also acknowledge the basal lepidosauran clade that defies the otherwise general trend you describe.

Figure adjusted

Style:

L. 66. Often using n-dash instead of comma's

This passage already revised

L. 64. Example of references clustered at the end of complex sentence where many sub-statements require specific references.

That specific example is suitable – Kuo and Ruina converge on the same principles. A later series covering force orientations is now split into substatements in the expanded history section.

L. 68. Broadly axial limb loading could be expressed clearer.

'Axially loaded limbs' description revised:

legs with force vectors passing along the line of the leg, close to the joint centres and broadly towards the centre of mass

L. 77 Reword: step periods that in a relative sense are very long?

Reworded to avoid possibility of 'long step' rather than 'long step duration' interpretation:

despite strides or relatively very long duration

L. 87. Either "a walking alligator" or "walking alligators" - same for tortoise.

pluralised

L. 176,389: moment arms can be great or large, not high.

'high' becomes 'large'

Sentence starting L. 187. pretty intense, could you please break it up?

Split in passage that also expands the who-did-what referencing.

L. 190. I think the meaning of early-skewed would be clearer if more words were added to the sentence. Then, the meaning of "late skew" in L. 193 follows naturally.

Expanded, along with splitting of sentence

L. 190. The meaning of "accounts" is not particularly clear here - I thought you were referring to previous studies as "accounts".

Changed to 'potential explanations'

L. 195-6. Maybe clearer as "With a limb loading that is more vertical than axial"?

Better. Thanks. Now:

Instead, it points to pitch avoidance with a limb loading that is more vertical than purely axial

L. 342: replace "they" with "these costs".

changed

L 354 with respect to the foot and located one leg length...4

changed

L 384. reword: works to work contributions?

changed

L386. replace the stance with either "the stance phase" or "stance"

'stance' deleted: of course it is only over stance.

Appendix B

Responses

The only query I have is from Board Member 1 concerning supplementary files. I do need help here: as Proc R Soc B cannot host compressed (.zip) folders, the code and outputs must instead be hosted by FigShare. I am not sure about this process.

In this round I submit 'ESM_Main' as the electronic supplementary information for the paper (a Word document). It would be nice if this could link to the Figshare site, but I don't know how to do this yet.

Thanks,

Jim.